# Anti-Inflammatory Salidroside Delivery from Chitin Hydrogels for NIR-II Image-Guided Therapy of Atopic Dermatitis

**DOI:** 10.3390/jfb14030150

**Published:** 2023-03-08

**Authors:** Shengnan He, Fang Xie, Wuyue Su, Haibin Luo, Deliang Chen, Jie Cai, Xuechuan Hong

**Affiliations:** 1State Key Laboratory of Virology, College of Science, Research Center for Ecology, Laboratory of Extreme Environmental Biological Resources and Adaptive Evolution, Medical College, Tibet University, Lhasa 850000, China; 2Hubei Engineering Centre of Natural Polymers-Based Medical Materials, College of Chemistry & Molecular Sciences, Wuhan University, Wuhan 430072, China; 3Key Laboratory of Tropical Biological Resources of Ministry of Education, School of Pharmaceutical Sciences, Hainan University, Haikou 570228, China; 4Jiangxi Key Laboratory of Organo-Pharmaceutical Chemistry, Chemistry and Chemical Engineering College, Gannan Normal University, Ganzhou 341000, China; 5Institute of Hepatobiliary Diseases, Zhongnan Hospital of Wuhan University, Wuhan 430071, China; 6Wuhan University Shenzhen Research Institute, Shenzhen 518057, China

**Keywords:** the quaternized β-chitin/dextran hydrogel, imaging in the second near-infrared window (NIR-II), atopic dermatitis, *Rhodiola rosea*, treatment

## Abstract

Atopic dermatitis (AD) is the most common heterogeneous skin disease. Currently, effective primary prevention approaches that hamper the occurrence of mild to moderate AD have not been reported. In this work, the quaternized β-chitin dextran (QCOD) hydrogel was adopted as a topical carrier system for topical and transdermal delivery of salidroside for the first time. The cumulative release value of salidroside reached ~82% after 72 h at pH 7.4, while in vitro drug release experiments proved that QCOD@Sal (QCOD@Salidroside) has a good, sustained release effect, and the effect of QCOD@Sal on atopic dermatitis mice was further investigated. QCOD@Sal could promote skin repair or AD by modulating inflammatory factors TNF-α and IL-6 without skin irritation. The present study also evaluated NIR-II image-guided therapy (NIR-II, 1000–1700 nm) of AD using QCOD@Sal. The treatment process of AD was monitored in real-time, and the extent of skin lesions and immune factors were correlated with the NIR-II fluorescence signals. These attractive results provide a new perspective for designing NIR-II probes for NIR-II imaging and image-guided therapy with QCOD@Sal.

## 1. Introduction

Atopic dermatitis (AD) is one of the common clinical chronic diseases with intense itching [1,2,3]. AD is a genetic predisposition, which is difficult to cure and relapses with diseases, such as allergic rhinitis and asthma, which severely influences the patient’s quality of life [4,5]. The pathophysiology of AD is attributed to a number of factors, such as inflammation, T cell infiltration, IgE-mediated sensitization, and neuroinflammation [3,6]. The main goals of AD therapy are to reduce skin inflammation and dysbiosis, improve the skin barrier, and avoid relevant disease triggers [7]. Current treatments for AD include anti-inflammatory agents such as corticosteroids, immunosuppressants, topical emollients, and phototherapy [8]. Currently, the effective primary prevention approaches that hamper the occurrence of mild to moderate AD have not been reported. Thus, developing targeted small-molecule drugs and biological therapies for moderate-to-severe AD diseases is in high demand.

The progression and severity of AD could be assessed by the Scoring Atopic Dermatitis (SCORAD) index or Eczema Area and Severity Index (EASI) [9]. However, these visual assessments are semi-quantitative and require experience. Hence, it is very desirable to have a non-invasive assessment during the therapeutic intervention and clinical trials. Recently, Raman confocal microspectroscopy (RCM) and optical coherence tomography (OCT) were used to assess the structural and functional abnormalities of AD in vivo [10,11]. Fluorescent imaging in the second near-infrared window (NIR-II, 1000–1700 nm) [12,13,14], first introduced in 2009, is an emerging imaging technique that offers non-invasive and higher-resolution images [12,15,16,17,18]. NIR-II imaging provides deep-seated anatomical and physiological imaging with micron-level resolution at the millimeter depth [18,19,20,21]. With these resolutions, the vascular transformation and drug efficacy assessment of AD could be detected by NIR-II imaging [22].

*Rhodiola rosea* is a perennial herb that originated in southwest China and the Himalayan Mountains [23,24]. Salidroside is the main active component of *Rhodiola rosea*. Salidroside is soluble in water and a very weakly acidic compound that has pharmacological effects against cerebral ischemia/reperfusion [25,26], renal fibrosis [27], nonalcoholic fatty liver, and myocardial injury [28]. It also could alleviate lipid accumulation and inflammatory responses in primary hepatocytes after palmitic acid/oleic acid stimulation [29,30,31,32,33]. In addition, salidroside can effectively prevent high-fat/high-cholesterol-diet-induced NASH (non-alcoholic steatohepatitis) progression by regulating glucose metabolism dysregulation [34,35], insulin resistance, lipid accumulation, inflammation, and fibrosis. In vivo and in vitro experiments have also demonstrated that salidroside could promote AMPK, NF-κB, IRF1/USF1, and KLF4/eNOS signaling pathways during the course of salidroside therapy [36,37]. However, there is still a lack of visual imaging evidence and the relevant mechanism from the image-guided salidroside therapy of AD.

Hydrogels, consisting of swellable networks of natural or synthetic polymers with high load-bearing capacity, have been widely used as tissue engineering scaffolds or delivery vehicles for therapeutic agents [38]. Common natural polymers for hydrogel synthesis are alginate, chitin/chitosan, gelatin, agarose, hyaluronic acids, cellulose, and other components [39,40,41]. Chitin, one of the most abundant amino polysaccharides in nature, with excellent biocompatibility, biodegradability, and bioactivity, has been extensively exploited in the last decades [42,43]. Nevertheless, the applications and developments are still limited due to their poor solubility and physiologically inertness. Recently, we have reported a green and efficient KOH/urea aqueous solution with which to construct high-strength chitin and chitosan hydrogels, films, and fibers [44,45,46,47,48,49,50]. The amazing results inspired Cai et al. to develop novel self-healing QCOD hydrogels based on β-chitin under physiological conditions by dynamic Schiff base linkage between the amino group of quaternized β-chitin (QC) and the aldehyde group of oxidized dextran (OD), while QCOD hydrogels have demonstrated excellent biocompatibility, capacity, and potential to be a depot for sustained release [51].

In this paper, the quaternized β-chitin/dextran (QCOD) hydrogel containing salidroside was designed, synthesized, and characterized as a topical carrier system for topical and transdermal delivery. The therapeutic mechanism of QCOD@Sal against AD was then systematically investigated. Finally, a NIR-II fluorophore HLA4P [21] for high-resolution NIR-II fluorescence imaging and NIR-II image-guided therapy of AD, using QCOD@Sal was studied. The treatment process of AD was dynamically monitored in real-time. These attractive results provide a new perspective for designing controlled drug delivery through the skin and improving the flexibility and therapeutic efficiency of AD.

## 2. Materials and Methods

### 2.1. Materials and Reagents

A 4:1 ratio of acetone (Bidepharm, Shanghai, China) and olive oil (Aladdin Biochemistry, Shanghai, China) was used to dissolve 1-fluoro-2,4-dinitrobenzene (DNFB, Tokyo Chemical Industry, Tokyo, Japan) [52]. The resulting DNFB solution was used in an animal model of AD [52,53,54,55]. The positive drug, dexamethasone acetate cream (Sanjiu Medical & Pharmaceutical Co., Guangzhou, China). All other chemicals and solvents were analytical grade. Squid pens were purchased from Zhejiang Jinke Company (Taizhou, China), and β-chitin was purified from squid pens according to our previous study [49]. Dextran, 2,3-epoxypropyltrimethylamonium chloride (EPTMAC) was purchased from Shanghai Chemical Reagent (Shanghai, China). Penicillin–streptomycin, RPMI-1640, and trypsin were purchased from Biological Industries (Shanghai, China), and fetal bovine serum (FBS) and Dulbecco’s phosphate-buffered saline (PBS) were purchased from Hyclone^TM^ (Shanghai, China). The 3-(4,5-Dimethylthiazol-2-yl)-2,5-diphenyl tetrazolium bromide (MTT) and the Calcein-AM/propidium iodide (PI) double stain kits were purchased from Shanghai Yusheng Biological Company (Shanghai, China).

### 2.2. Synthesis of Quaternized Chitin (QC) and Oxidized Dextran (OD)

QC was synthesized according to the previous work, with a minor modification (Appendix A) [56]. A total of 1.0 g of purified β-chitin powder was dispersed into a mixed solution of KOH/urea/H_2_O in a ratio of 20:4:75 (*w*/*w*/*w*) to obtain a suspension (1 wt%). Then, it was stirred at −30 °C for ~30 min to form a clear and viscous chitin solution. EPTMAC (EPTMAC: *N* -acetyl- D -glucosamine units (GlcNAcU) = 8:1 mol/mol) was added to the chitin solution and stirred at 40 °C for an additional 24 h. The mixture was neutralized with hydrochloric acid and the resulting quaternized chitin derivatives were dialyzed in distilled water using a dialysis membrane for more than 7 days. The final product was freeze-dried and stored in a moisture-free desiccator prior to use. OD was prepared according to previous work [51]. Briefly, an aqueous dextran solution (2.0 g/200 mL distilled water) was oxidized with 1.0 g of NaIO_4_ at 25 °C for 2.5 h. The reaction was quenched by adding 0.5 mL of ethylene glycol, and the mixture was stirred for an additional 1 h. The above solution was, then, dialyzed in distilled water and freeze-dried.

### 2.3. Preparation and Characterization of QCOD Hydrogels

QC was dissolved in a phosphate buffer solution (PBS, pH 7.4) or physiological saline at a concentration of 1.0% (*w*/*v*). OD was dissolved in a phosphate buffer solution (PBS, pH 7.4) or physiological saline at a concentration of 2.0% (*w*/*v*). The QCOD hydrogels were obtained by mixing equal volumes of the QC and OD solutions at 25 °C (Appendix A). The content of aldehyde groups was determined by hydroxylamine hydrochloride titration [57], and the degree of oxidation was determined to be 1.10 mmol (CHO)/g. The degree of quaternization (*DQ*) was determined by conductometric titration of the Cl^−^ ion concentration using a standard AgNO_3_ solution in water [58]. The degree of deacetylation (*DD*) of QC was determined by the two-abrupt-change potentiometric titration method [59]. The weight-average molecular weight (*M*_w_) was measured by size exclusion chromatography (SEC) with multi-angle static light scattering (MALS) (DAWN HELLEOS-II, Wyatt, USA, a He–Ne laser, λ = 663.4 nm), a differential refractometer (RI) (Opitilab T-rEX, λ = 658.0 nm), and a capillary viscosity detector (ViscoStar-II), as previously described, with a 0.1 M NH_4_Ac/HAc aqueous solution as the eluent and at a flow rate of 0.5 mL/min. Nuclear magnetic resonance (NMR) measurements were carried out on a Varian INOVA-600 spectrometer in the proton noise decoupling mode with deuterated water (D_2_O) as the solvent.

The compressive strength of the hydrogels was determined on a Discovery HR-2 Rheometer (TA Instruments, Framingham, MA, USA). The specific test methods were as follows: The QC and OD solutions were rapidly mixed and placed on the rheometer plate. The upper plate was immediately lowered to a measurement gap size of 0.3 mm. The rheology properties of QCOD hydrogels were, then, measured using a Discovery HR-2 Rheometer (TA Instruments, USA) with the steel parallel-plate geometry (40 mm diameter). Strain amplitudes were determined and set to 1% to ensure that all measurements were performed within the linear viscoelastic regime. Time-sweep tests were performed at 37 °C using a constant angular frequency of 10 rad/s to record the storage modulus (*G′*), and loss modulus (*G″*) versus time for the QCOD hydrogels. Frequency-(*ω*) sweep tests from 0.1 to 100 rad/s were performed in the linear viscoelastic region of QCOD hydrogels at 37 °C after 1 h of pre-gelation. The strain amplitude of the oscillations was switched from a low level (*γ* = 1% at each interval of 2 min) to a high level (γ = 350% at each interval of 2 min), and three cycles were performed in the experiments.

The moisture retention experiment of the hydrogels was carried out in a closed desiccator containing desiccant, using pure water and dexamethasone ointment as the negative control and positive control, respectively, to measure the time-related water loss curve of the QCOD hydrogels. A certain mass (*W_0_*) of the hydrogels, water, or dexamethasone ointment was taken and placed in the desiccator; then, the samples were weighed at different time intervals (*W_t_*), and the water loss rate was calculated at different times using the following equation:Water loss rate (%) = *(W_0_* − *W_t_*)/*W_0_* × 100

### 2.4. Preparation of QCOD@Sal Hydrogels

QC was dissolved in phosphate buffer solution (PBS, pH 7.4) or physiological saline at a concentration of 1.0% (*w*/*v*). OD and salidroside were dissolved in phosphate buffer solution (PBS, pH 7.4) or physiological saline at a concentration of 2.0% (*w*/*v*). The OD solution (0.25 mL) containing salidroside and QC solution (0.25 mL) were mixed uniformly at 25 °C before it was settled for 2 min to obtain the salidroside-loaded hydrogel (0.5 mL). In the cell and animal experiments, the concentration of salidroside was 0.25%.

### 2.5. The Controlled Release of Salidroside In Vitro

The standard curve of salidroside aqueous solution at 275 nm was first determined (y = 0.0477x + 0.00607, y: the UV absorbance of salidroside aqueous solution at 275 nm, x: the concentration of the salidroside aqueous solution, µg/mL). Firstly, different concentrations of the salidroside aqueous solution (9.375, 12.5, 15.625, 25, 31.25, 50, 62.5, 83.33, 125, and 250 μg/mL) were prepared, then, the absorbance was measured at 275 nm by UV spectrophotometer, and finally, the absorbance values and solution concentrations were used as the horizontal and vertical coordinates to make the curves and perform the standard curve (Appendix A).

The OD solution (0.25 mL), containing salidroside (5 mg/mL), and the QC solution (0.25 mL) were mixed uniformly at 25 °C, and then, it was settled for 2 min to obtain salidroside-loaded hydrogels (0.5 mL). Salidroside-loaded hydrogels (0.5 mL) were immersed in 2 mL of PBS at pH 5.0 or 7.4 at 37 °C. The concentration of salidroside in all hydrogels was 2.5 mg/mL in 0.5 mL, with three parallel sets of samples at each time point. Furthermore, at predetermined time intervals (1 h, 2 h, 4 h, 6 h, 8 h, 12 h, 24 h, 36 h, and 72 h), 1.2 mL of the incubated solution was removed and the UV absorbance was recorded at 275 nm. The UV absorbance at 275 nm was calculated by bringing in the standard curve formula to determine the concentration of salidroside (A*_x_*). The salidroside aqueous solution (2.5 mg/mL, 0.5 mL), immersed in 2 mL of PBS at pH 5.0 and 7.4, were used as the control, while the UV absorbance of PBS at 275 nm, at predetermined time intervals, was recorded to determine the concentration of salidroside (A*_1_*). The controlled release of salidroside in the hydrogel was determined by the following formula:Release of salidroside (%) = A*_X_*/A*_1_ *× 100

### 2.6. MTT Cytotoxicity Assay

Mouse fibroblast (L929) cells were propagated in RPMI-1640 with 10% FBS and 1% penicillin-streptomycin. To evaluate the cytotoxicity of the QCOD hydrogels and QCOD@Sal hydrogels, the cell suspension (~5 × 10^3^ cells/well) was seeded in 96-well tissue culture plates and incubated in 200 μL of RPMI-1640 for 24 h. The cell culture medium was then replaced with fresh media containing QCOD hydrogels (10 μL) or QCOD@Sal hydrogels (10 μL). The MTT assay was used to assess the cell viability after 1, 2, and 3 days of incubation. After 1, 2, and 3 days of incubation, 10 µL of MTT (5 mg/mL) solution was added to each well and incubated for an additional 4 h. The supernatant was discarded, and dimethyl sulfoxide (DMSO) was added at 200 µL/well and shaken for ~10 min until the blue methane crystals were dissolved. The cells containing only medium were used as the control. The absorbance at 570 nm was measured and cell viability was calculated using the following formula:The cell viability (%) = OD_570sample_/OD_570control_ × 100

The cell viability was also characterized using a live/dead assay kit (Calcein AM/PI). Calcein AM is a dye that can enter into the cell and be hydrolyzed by esterase to yield green fluorescence (excitation at 490 nm and emission at 515 nm). The PI dye is non-cell permeable and can only stain dead cells with red fluorescence when interacting with cellular DNA (excitation wavelength 535 nm, emission wavelength 617 nm). After 1, 2, and 3 days of co-culturing cells (1 × 10^4^ cells/well) and hydrogels (50 μL) in 24-well tissue culture plates, the cell-seeded samples were washed thoroughly with PBS and incubated in a standard working dye solution of Calcein AM/PI (5 µL) and PBS (200 µL) for 30 min. After again washing with PBS, the sample images were obtained using a fluorescence microscope (Leica, Wetzlar, Germany).

### 2.7. Hemolysis Assay

The whole blood was collected from mice and centrifuged at 1000 rpm for 10 min. After centrifugation, the supernatant was removed and washed with PBS (3 × 10 mL) until it was no longer red and cloudy. Deposited red blood cells were diluted with PBS to obtain a suspension of red blood cells at a concentration of 5% *v*/*v*. The hydrogel was mixed with red blood cells (1:1). The groups were divided into the negative control group, the positive control group, the QCOD group, and the QCOD@Sal group. The negative control group was prepared from 200 microliters of PBS buffer and 200 microliters of red blood cell suspension. The positive control group consisted of 200 microliters of primary water and 200 microliters of red blood cell suspension. The QCOD group was composed of 200 microliters of QCOD hydrogel and 200 microliters of red blood cell suspension. A total of 200 microliters of QCOD@Sal was mixed with 200 microliters of red blood cells, then, the supernatant was centrifuged and placed in a 96-well plate, 100 microliters per well, and each sample was re-welled 4 times. The absorbance was measured at a wavelength of 540 nm. The hemolysis rate of the hydrogel was calculated using the following equation:Hemolysis rate (%) = (A_experimental group_ − A_negative control group_)/(A_positive control_ − A_negative control group_) × 100%

### 2.8. Animal Models and In Vivo Therapy

For the experimental animals, six-week-old male BALB/c mice (Laboratory Animal Center, China Three Gorges University, Animal ethics: WAEF2021-0016) were purchased and kept in the individually ventilated cages (IVC) system in the laboratory animal center of Wuhan University, maintaining sufficient drinking water and feed. All animal experiments were approved by the Wuhan University Animal Ethics Committee and conducted according to the ARRIVE guidelines. A total of 30 male BALB/c mice (age, 6–8 weeks; weight, 18–20 g; n = 6/group) were purchased from China Three Gorges University and acclimatized for one week after being transferred to the individually ventilated cage system. The hair on the back of the mice was removed over an area of 3 × 2 cm using a depilatory paste one day before the following experiment. On days 1, 2, 3, and 4, the mice were sensitized with a 0.5% DNFB (acetone:olive oil = 4:1, 100 μL) on the shaved areas. On day 9, the mice were sensitized with 0.2% DNFB (100 μL) to induce an AD-like phenotype. In the control group, the acetone and olive oil mixture (4:1, 100 μL) was applied to the back area of mice at the same time. The mice were photographed every two days and the skin severity was evaluated according to SCORing atopic dermatitis. The erythema/deepening color, edema/popularity, exudation/crusting, exfoliation, mossy/itchy rash, and dryness were evaluated. The maximum score for each item was 3 points, and the maximum score for each group was 18 points. After the successful modeling, all mice were divided into five groups, the normal group (normal), the control group treated with saline, the QCOD group treated with QCOD, the QCOD@Sal group treated with QCOD@Sal, and the DEX group treated with 0.1 mg/kg dexamethasone (DEX). For the treatment, QCOD@Sal was daubed every day (0.25% per application) [60], and the daubed position was the same. At the end of the study, all the animals were anesthetized with pentobarbital sodium. Blood was collected and skin samples were collected for analysis.

### 2.9. Antioxidant Activity of the Hydrogel

The antioxidant efficiency of QCOD@Sal hydrogels was evaluated by scavenging the stable 2, 2-diphenyl-1-picrylhydrazyl (DPPH) free radicals (Appendix A).

### 2.10. NIR-II Imaging and Image-Guided Therapy of AD

All NIR-II fluorescent images were collected using the NIR-II imaging system (Suzhou NIR-Optics Technologies Co., Ltd, Suzhou, China, 808 nm, 3.5 mW cm^−2^_,_ 1000 nm LP). The mice were anesthetized by pentobarbital sodium and mounted at a height of 12 cm during NIR-II imaging of AD. There were five groups: the AD group, QCOD@Sal group, QCOD group, DEX group, and NOR group. Each group had six mice. After the treatment of AD with QCOD, QCOD@Sal, or DEX, the NIR-II probe HLA4P was injected into the tail vein (Appendix A). Li [35] provided the synthetic route of the probe. Then, the back skin damage was visualized in real-time on days 5, 10, and 15. The therapeutic effect was reflected according to the NIR-II fluorescence intensity (FL. intensity) (Appendix A).

### 2.11. Measurement of IL-6 and TNF-α Release and Blood Routine Examination

Blood was collected from each mouse at the end of the experiment. The whole blood samples should be placed at room temperature for 2 h or 4 °C overnight and then, separated at 3000 rpm, 2–8 °C, for 15 min. The supernatant was taken for immediate detection. ELISA kit (Servicebio, Wuhan, China) was used for the analysis of IL-6 and TNF-α release, and the operation was carried out according to the instructions. The absorbance value was read at 450 nm with a microplate reader (Spark, Männedorf, Switzerland). A blood routine examination was performed using the automatic blood cell analyzer (BC-2800VET, Mindray Animal Medical, Shenzhen, China).

### 2.12. Histological Analysis

To evaluate the histology of skin tissue, skin tissues from BALB/c mice were taken and fixed with a 4% paraformaldehyde solution. Tissue samples were stained by H&E, TUNEL, and ROS.

The procedure for H&E staining is as follows: After the treatment of the mice, the skin, about 1 cm × 1 cm on the back of the mice, was dissected with surgical scissors, and the removed skin was fixed in 4% paraformaldehyde solution. The detailed procedure for H&E staining: The sections were placed into a staining cylinder and stained with hematoxylin staining solution for 10 min. The cut was removed and washed with the staining cup water until the sections were colorless. The cells were differentiated with 1% hydrochloric ethanol differentiation solution for 3 s, followed by rinsing with water for 5 min. Then, the sections were stained with eosin for 5 min, followed by immersion in water for 2 min. The sections were successively dehydrated with 80% ethanol, 90% ethanol, 95% ethanol, and 100% ethanol for 2 min. Finally, the slices were placed in xylene I for 2 min and placed in xylene II for 2 min. After drying the xylene on the back side, the neutral resin was added to the front side to seal the slices. The typical histopathological changes of the tissues were observed under a 400 light microscope (ECLIPSE Ci, Nikon, Tokyo, Japan) after staining with an H&E staining kit (Pinofil Biotechnology Co., Ltd., Pinofil, Wuhan, China).

The TUNEL staining process was as follows: The frozen tissues were dehydrated with sucrose and then embedded by OCT. The embedded blocks were frozen and sliced. Before the experiment, the slices were taken out and placed on the staining rack for 15 min with a fixed solution, and then, treated at 37 °C for 5 min with protease K (in situ Cell Death Detection Kit, POD). Then, the slices were incubated at 37 °C for 2 h. After DAPI staining (Beyotime C1002, Beyotime, Shanghai, China), the slices were imaged under a fluorescence microscope (ECLIPSE Ci-L, Nikon, Tokyo, Japan).

ROS determination: The frozen sections were rewarmed, and the pen circles were organized. The slices were stained with DHE (Sigma D7008) and incubated at 37 °C for 30 min in the dark condition. Then, the slices were stained with DAPI dye solution (Beyotime C1002, Beyotime, Shanghai, China) for 10 min in the dark and sealed. The sections were observed under an inverted fluorescence microscope (ECLIPSE Ci-L, Nikon, Tokyo, Japan) and the images were collected.

### 2.13. Statistical Analysis

All statistical data were performed using GraphPad Prism 9.0 and presented as means ± standard deviation. Data are presented as mean ± S.D. Comparisons of means of ≥3 groups were performed by analysis of variance (ANOVA) and the existence of individual differences, in case of significant F values at ANOVA, were assessed by multiple contrasts. Values of *p** < 0.05 were considered statistically significant, *p*** < 0.01 was considered statistically significant, and *p**** < 0.001 was considered extremely significant.

## 3. Results and Discussions

### 3.1. Preparation and Characterization of QCOD and QCOD@Sal Hydrogels

As shown in Appendix A, the quaternization of chitin was first carried out at 40 °C with a degree of deacetylation (*DD*) of 35% and a weight-average molecular weight (*M_w_*) of 1.6 × 10^5^ g/mol. The quaternized chitin (QC) was homogeneously synthesized in an aqueous KOH/urea solution through a high-efficiency, energy-saving, green pathway (Appendix A). On the other hand, the oxidized dextran (OD) was obtained by oxidation reaction. The structures of QC and OD were characterized by ^1^H NMR analysis and potentiometric titration (Appendix A). A 2.69 ppm chemical shift was the C2 proton peak of the free amino group of the quaternary chitin. The results indicated that a large number of amino groups were generated, and the quaternary ammonium reaction of chitin was successfully deacetylated under a condition of strong alkali and high temperature. The results were consistent with the results of the potentiometric titration. In addition, the new signal of oxidized dextran at 4–6 ppm was the hemiacetal proton peak, and 9.0 ppm was the proton signal of the free aldehyde group on the oxidized dextran. The QCOD hydrogels were quickly prepared by crosslinking QC and OD under physiological conditions through the dynamic Schiff base linkage formation (Figure 1b). The scanning electron microscopy (SEM) image showed that QCOD hydrogels possessed a 3D network consisting of polysaccharide nanofibers, indicating the encapsulating and delivering capacity of QCOD hydrogels (Figure 1c).

Then, the rheology properties of QCOD hydrogels were determined by a rheometer. As shown in Figure 1d, the gelation process of QCOD hydrogels was further monitored by dynamic time-sweep rheology experiments. In the early stage, storage modulus (*G′*) and loss modulus (*G″*) increased rapidly. A rapid crosslinking and gelation process was shown when *G′* exceeded *G″*, exhibiting solid-like elastic gel properties. The QCOD hydrogels were further subjected to frequency-sweep tests by dynamic rheology at 37 °C, and the storage modulus of the hydrogel networks was 1800 Pa (Figure 1e). The cumulative release value of salidroside reached ~82% after 72 h at pH 7.4, while the maximum release value reached ~95% after 72 h, at a relatively acidic pH value of 5.0. The acid-sensitive covalent linkages of the QCOD hydrogel networks were closely associated with decreased pH, resulting in increased release of salidroside from QCOD hydrogels under acidic conditions. In addition, QCOD@Sal hydrogels have certain scavenging abilities for free radicals, and the scavenging rate increased gradually with the increase in concentration (Appendix A). The excellent water retention of QCOD was observed due to its unique 3D network consisting of polysaccharide nanofibers. The experimental results also showed that at 144 h, the water loss rates of pure water and low dexamethasone ointment groups were ~65% and ~57%, respectively, while the water loss rate of the hydrogel group was ~28%, indicating that the hydrogel had better water retention. The experimental results showed that at 144 h, the water loss rates of pure water and low dexamethasone ointment groups were ~65% and ~57%, respectively, while the water loss rate of the hydrogel group was ~28%, indicating that the hydrogel had better water retention (Figure 1f). Next, the anti-inflammatory drug, salidroside was encapsulated into QCOD hydrogels, and the pH-responsive drug release behavior was studied at 37 °C (Figure 1f). As shown in Appendix A, QCOD@Sal has the ability to scavenge free radicals, and the antioxidant capacity was increased with the increase in concentration. The above results suggest that QCOD hydrogels can be used as a carrier for the sustained delivery of anti-inflammatory drugs.

The dynamic imine bonds of QCOD hydrogels resulted in shear-thinning and self-healing capabilities. The viscosity of QCOD hydrogels decreased with the increasing shear rate, which allows the pre-formed hydrogel to encapsulate and deliver salidroside in vivo (Figure 2a). QCOD hydrogels can be pushed out of the WHU letter by a syringe. Furthermore, the strain sweep measurements were performed (Figure 2c). The strains at the intersection of *G′* and *G″* were ~280%, indicating a critical state of colloid and solution. When the shear strains were greater than 350%, QCOD hydrogels exhibited a sol-gel transition due to the dynamic imine bond breaking and polymeric chain reorientation within the hydrogel network.

The self-healing properties of QCOD hydrogels were studied by alternating high and low strain (between 1% and 500% strain) scanning modes (Figure 2d). As the amplitude of the oscillatory force increased from low to high (1% strain to the 500% strain), the *G′* value immediately dropped from 1800 Pa to 190 Pa, and the storage modulus was lower than *G″*. When the strain returned to 1%, the *G′* and *G″* values rapidly returned to their initial values. The rapid recovery of hydrogen and imine bonds within the reversible hydrogel network may cause reversible recovery behavior, indicating that QCOD hydrogels can self-heal quickly and efficiently, allowing them to be used as drug carriers.

The biocompatibility of QCOD and QCOD@Sal hydrogels was then evaluated by the MTT assay and hemolysis assay. The mouse fibroblast cells L929 were used to investigate the cell viabilities of the QCOD and QCOD@Sal hydrogels (10 μL hydrogel/200 μL medium) at different culture times (on days 1, 2, and 3). As shown in Figure 3a, compared with the control group, the survival rate of the L929 cells co-cultured with QCOD and QCOD@Sal on days 1, 2, and 3 was >85%, which shows excellent cytocompatibility. In addition, the activity and morphology of the L929 cells were further assessed by cytofluorimetric staining assays. Almost all L929 cells emitted green fluorescence, and most of the cells showed a full oval or shuttle shape (Figure 3b). In addition, the hemolysis values of QCOD and QCOD@Sal hydrogels on red blood cells were 1.1% and 4.9%, respectively, indicating that QCOD and QCOD@Sal hydrogels showed excellent red blood cell compatibility (Figure 3c). The excellent biocompatibility of QCOD@Sal was related to its composition. The material selection was made of chitin, one of the most abundant amino polysaccharides in nature with excellent biocompatibility and biodegradability, while its bioactivity has been intensively exploited in the last decades.

### 3.2. NIR-II Imaging and Image-Guided Therapy of DNFB-Induced AD Mice

We further explored the potential of HLA4P [35] (200 μM) for NIR-II fluorescence imaging of DNFB-induced AD mice (n = 6), with or without treatment, in vivo. In vivo NIR-II imaging was carried out 3, 5, 10, and 15 days after injecting HLA4P [35], under 808 nm excitation (3.5 mW cm^−2^). The body weight changes were also recorded. As shown in Figure 4b, no significant differences were observed in the body weights among the treatment group (Figure 4b). Among them, the QCOD@Sal treatment group showed the best therapeutic effect on the dermatitis of DNFB-induced AD mice. It can be seen that QCOD@Sal enabled the sustained release of salidroside and enhanced the wound closure rate on day 6, while the wound was basically healed on day 15 (Figure 4c). Thus, QCOD@Sal hydrogels not only moisturized the damaged skin, yet also provided a good anti-inflammatory effect in the presence of salidroside. In the DEX treatment group, the eschar began to show on day 15, while the damaged skin was accompanied by obvious dryness, which was inferior to the QCOD@Sal treatment group. As illustrated in Figure 4d, the AD symptoms were plainly visible against the background at 120 h after injection of HLA4P. Li [35] provided the synthetic route of the probe. The probe accumulation peaked at 120 h after injection without any treatment (Figure 4d). The fluorescence signal of the QCOD@Sal treatment group was much weaker at all timepoints compared to the DEX and QCOD treatment groups (Appendix A). During the treatment, the SCORAD scores were performed on the back skin damage of the mice on days 8, 12, and 14. Analysis of the data revealed that the QCOD@Sal treatment group was more effective than the QCOD and DEX treatment groups. The back skin dermatitis of mice improved as the treatment progressed and the QCOD@Sal treatment group had a significant difference in the dermatitis score compared to the DEX group (*p**** < 0.001) (Figure 4e).

### 3.3. Mechanism of QCOD@Sal for Ameliorating AD

To further verify the mechanism underlying the QCOD@Sal therapeutic effect in the AD treatment, we performed H&E staining, IHC-Fr staining, TUNEL staining, whole blood indicators, ROS levels, and IL-6 and TNF-α indicators. The data indicated that the expressions of central granulocytes, monocytes, macrophages, and lymphocytes in the control group were significantly increased, while they were significantly decreased in the following order: DEX, QCOD, and QCOD@Sal treatment groups (Figure 5a). TUNEL staining showed that apoptosis and inflammation were significantly elevated in the control group. The number of apoptotic cells following treatment with DEX was also significantly higher than that in the QCOD or QCOD@Sal treatment groups, yet still significantly lower than in the control group. Furthermore, the magnitude of the reduction in apoptosis after treatment with QCOD was slightly less than after QCOD@Sal treatment, which further confirmed the efficacy of the QCOD@Sal hydrogels in treating atopic dermatitis. A QCOD@Sal therapeutic effect in AD treatment was further characterized by the ROS (DHE) staining and showed that the ROS levels were in the order of the control group > DEX group > QCOD group > QCOD@Sal, with QCOD@Sal providing the most therapeutic efficiency for AD treatment (Figure 5a). The whole blood analysis of the AD mice without treatment was performed after orbital blood sampling on day 16, and elevated leukocytes, neutrophils, and lymphocytes were observed with an inflammatory response. The results also showed that leukocytes, lymphocytes, and neutrophils in the QCOD@Sal treatment group were significantly decreased relative to the DEX and QCOD treatment groups (*p**** < 0.001) (Figure 5b).

Considering the expression of the inflammatory factors on the wound surface, IL-6 and TNF-α were analyzed by ELISA on day 16, and the therapeutic effects of the QCOD@Sal group, QCOD group, and DEX group were evaluated. The results showed that the QCOD@Sal treatment group had the lowest IL-6 and TNF-α values, indicating less inflammation (Figure 5c).

## 4. Conclusions

In this work, we designed quaternized β-chitin/oxidized dextran hydrogels QCOD@Sal containing the traditional Tibetan medicine salidroside, which is in line with the concept of green chemistry and non-toxicity. QC and OD can generate hydrogels in situ under physiological conditions through dynamic imine bonding, and QCOD hydrogels have good drug encapsulation, sustained release, and moisturizing properties compared with other cutaneous drug delivery systems, including nanoparticles, ethosomes, and microneedles. QCOD hydrogels are convenient for drug delivery and belong to smear dosage forms, which do not pass through the liver and have no first-pass effects compared with oral dosage forms. QCOD@Sal hydrogels can be obtained by using QCOD hydrogels to encapsulate salidroside and moisturized the damaged skin and accelerated wound debridement during the treatment of atopic dermatitis. The results indicate that QCOD@Sal treated the DNFB-induced mouse model of AD by blocking the IL-6 and TNF-α signaling pathways. The extent of the skin lesions in the treatment process of AD was monitored in real-time by NIR-II fluorescence imaging for the first time. In future studies, the mechanisms underlying the effects of QCOD@Sal in the treatment of AD will be further determined. Our study evidenced that chitosan-related hydrogels QCOD are biocompatible materials that can be used as a carrier to improve the efficacy of Chinese herbal treatments. This encouraging potential enables natural derivate hydrogel manipulation and allows future integration of all medicines into one single treatment in a feasible and green way.

## Figures and Tables

**Figure 1 jfb-14-00150-f001:**
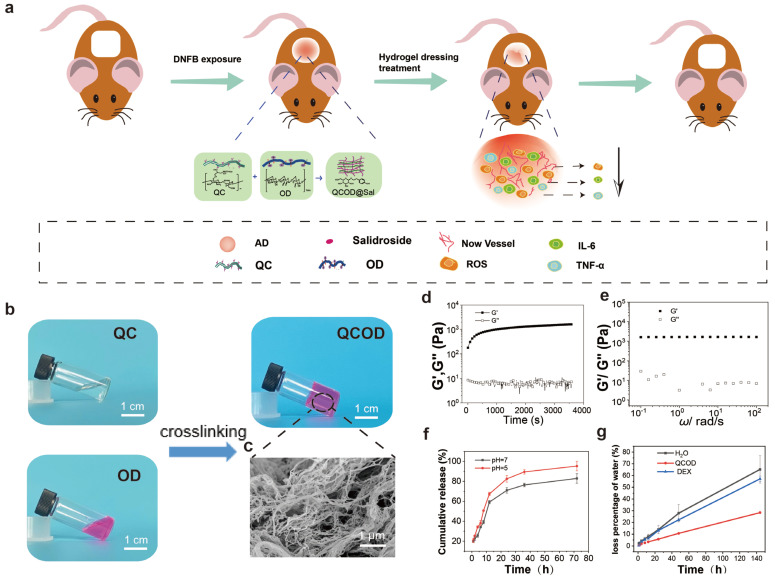
(**a**) A schematic picture of QCOD and QCOD@Sal hydrogels for the treatment of AD; (**b**) QCOD hydrogels can be obtained by mixing the QC solution and OD solution (stained with rhodamine b), and (**c**) the SEM image of QCOD hydrogels shows an interconnected polysaccharide nanofibrillar network structure; (**d**) storage modulus (*G′*) and loss modulus (*G″*) of QCOD hydrogels at different concentrations, at 37 °C, and as a function of time (s); (**e**) *G′* and *G″* of QCOD hydrogels as a function of frequency (*ω*); (**f**) in vitro drug release curves of QCOD hydrogels at pH = 7.4 and pH = 5.0 (n = 3); (**g**) water loss time curves of hydrogels in a closed desiccant containing desiccant.

**Figure 2 jfb-14-00150-f002:**
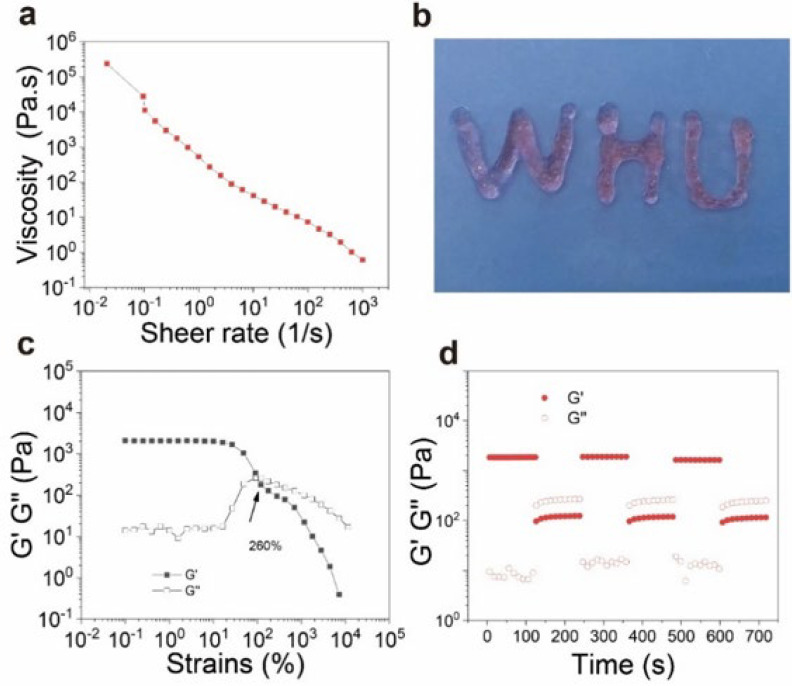
(**a**) The viscosity and shear-thinning behavior of QCOD hydrogels; (**b**) QCOD hydrogels can be pushed out of the WHU letter by a syringe; (**c**) the continuous step-strain measurements of QCOD hydrogels at 10 rad/s; (**d**) the strain interval of QCOD hydrogels was maintained at 120 s for the 1% strain and 120 s for the 500% strain.

**Figure 3 jfb-14-00150-f003:**
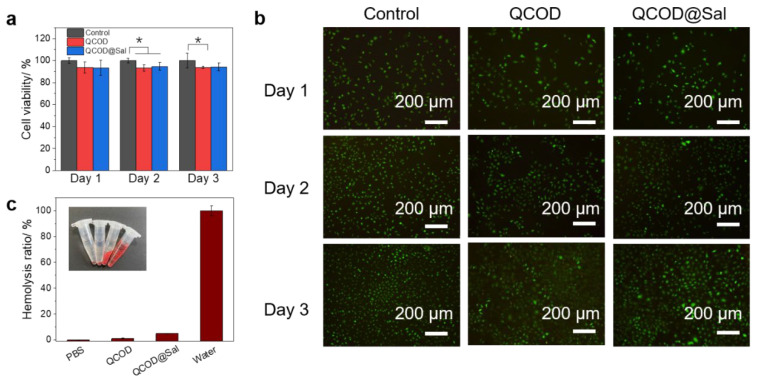
(**a**) The cell viability of L929 cells cultured with QCOD and QCOD@Sal hydrogels; (**b**) the fluorescence pictures of L929 cells cultured with QCOD and QCOD@Sal hydrogels on days 1, 2, and 3, respectively (dead cells: red and living cells: green); (**c**) the hemolysis values of red blood cells co-cultured with QCOD and QCOD@Sal hydrogels. *p** < 0.05.

**Figure 4 jfb-14-00150-f004:**
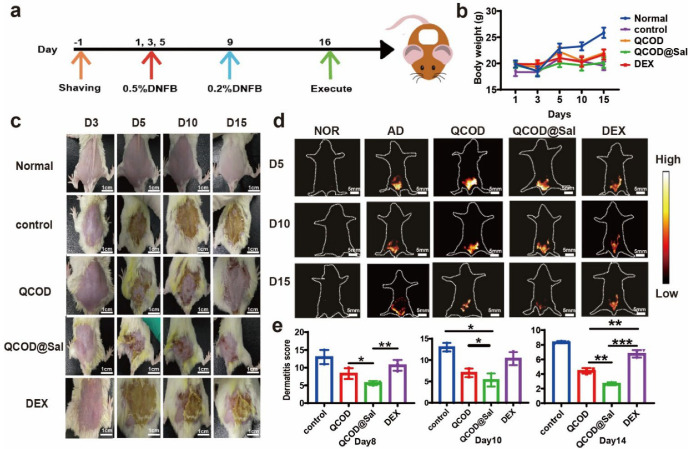
(**a**) A diagram of the dermatitis model. Dehairing was performed one day prior to modeling and induction was performed with 0.5% DNFB on days 1, 3, and 5. Excitation was performed with 0.2% DNFB on day 9. Drug treatment was performed 2 h after induction on day 3. Mice were executed by decortication on day 16; (**b**) mice body weight change graphs were examined on days 1, 3, 5, 10, and 15; (**c**) the skin lesions on the back of the mice were recorded with an iPhone 11 camera; (**d**) NIR-II fluorescence imaging was performed to monitor the dorsal skin lesions in the mice. NIR-II fluorescence imaging was performed on mice (Suzhou NIR-Optics Technologies Co., Ltd., 808 nm, 3.5 mW cm^−2^_,_ 1000 nm LP); (**e**) The scoring of the dorsal skin lesions in mice (n = 6). *p** < 0.05, *p*** < 0.01, and *p**** < 0.001.

**Figure 5 jfb-14-00150-f005:**
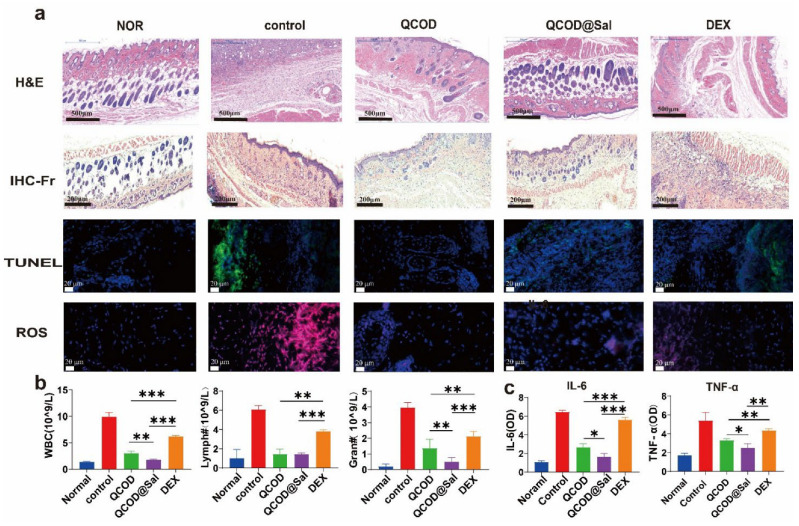
(**a**) The representative histopathological features of skin sections stained with H&E, IHC-Fr, TUNEL, and ROS content in tissues; (**b**) the analysis of whole blood of mice (n = 6), the contents of white blood cells, lymphocytes, and centrocytes were presented; (**c**) TNF-A and IL-6 were detected by ELISA. *p** < 0.05, *p*** < 0.01, and *p**** < 0.001.

## Data Availability

Data sharing not applicable No new data were created or analyzed in this study. Data sharing is not applicable to this article.

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
