# Peer review of "Anti-Inflammatory Salidroside Delivery from Chitin Hydrogels for NIR-II Image-Guided Therapy of Atopic Dermatitis"

_jfb, 2023, doi:10.3390/jfb14030150_

Round 1

Reviewer 1 Report

I recommend major revision;

  1. Hydrogels are generally designed as subcutaneous implants to release a controlled amount of drug over a long time period, so what is the authors' justification for the topical use of this carrier?
  2. In the MTT assay section, how did the authors sterilize the hydrogels?
  3. In line 191 what is the meaning of "standard working dye solution"?
  4. In the animal study section, is line 217 (DPPH assay) placed in a suitable section?
  5. TUNEL test should be evaluated for skin tissue.
  6. ROS should be evaluated in skin tissue.
  7. In the discussion section, the authors have not compared their results with other types of carriers such as ethosomes.
  8. The drug release should be evaluated via the Franz cell method.

Author Response

Thanks for your suggestion. Please see the attachment. 

Reviewer 2 Report

Authors presented extremely interesting research; however it is poorly described. A lot of information is missing, it is really difficult to follow the figures as the symbols of the figures are mistaken. The section “results and discussion” presents only results. Authors should answer questions: What does (the result) mean and why we obtain it. Moreover, obtained results should be discussed with other paper. Many information (both in methods and results) are just mentioned without proper explanation. 

1)    In my opinion the Introduction should start with the description of problem (AD) and followed by the solution.

2)    Why the current treatment of AD is insufficient (why authors are looking for new treatment)?

3)    In the section 2.3 a few methods were well described, however some of them (for example mechanical testing) lacks details. Please make the section 2.3 more consistent. 

4)    How the standard curve of salidroside dissolution was prepared, please provide the details. 

5)    Page 4, lines 169-172, how exactly the salidroside-loaded hydrogels were prepared, please provide the details (for example how long it was stirred, provide rpm, etc.)

6)    How many time intervals were performed in the release test? Did you have separate sample for each time point or after sample removal, fresh release medium was added to the hydrogel (cumulative release)? According to the graph cumulative release was performed, please clarify this. 

7)    Please provide the details about MTT assay. What did you do after sample incubation, how much MTT did you add, how did you dissolve the MTT, what wavelength did you use to measure the absorbance and what kind of plate reader you used?

8)    What is standard dye solution? Please provide details about the microscope and used lenses. 

9)    Please provide the details about H&E staining and used microscope and lenses.

10) Please give more details about HNMR results, authors should discuss chemical bonds.

11) Page 6 line 255, what does it mean “rheology test” ? Please be more precise. 

12) Page 6 line 256, please check if you refer to the proper Figure, shouldn’t be 1b instead of 1c?

13) Page 6 line 261 please refer to the figure.

14) Page 6 line 269 “Thus, QCOD hydrogels can improve dry skin and relieve dermatitis symptoms for a long time” this sentence is too general. How QCOD hydrogels can improve dry skin, by keeping water in the hydrogel? Please explain this phenomenon. 

15) Please check all the refers to the figure 1 in the paragraph 3, for example in the test authors referred to the figure 1d, while in the manuscript is 1g. 

16) On the figure 1g red line missing the standard deviation. 

17) Figure 3a lacks statistic.

18) How was the survival rate of mouse fibroblasts (cell viability) calculated? Authors should compare and discuss cell viability between the samples. 

19) Page 8 lines 326-328, please explain with details what authors can observe within fluorescent microscope? What does it mean normal morphology? It is really difficult to examine cells morphology within the image magnification authors presented in the manuscript. I would advise authors to add images with greater magnification if they want to discuss the morphology. Please check if the refer to figure 4c is necessary at that point of manuscript.

20) Hydrolysis assay is not well described. 

21) Page 9 lines 342-344, please explain changes in mice weight. 

22) Page 9 lines 348-350, “Thus, QCOD@sal hydrogels not only moisturized the damaged skin moist, bust also has good anti-inflammatory effect in the presence of salidroside.” Did authors check the moisturizing effect of hydrogels?

Author Response

(The authors gave the same response as above.)

Reviewer 3 Report

The study is interesting but requires a detailed revision regarding the following issues:

The document needs to be revised regarding typographical errors (line 237; BALA/c mice) and English grammar and syntax mistakes.

Title – The current title is not informative; the authors could improve it by including the main findings of the study;

Abstract – revise the abbreviations; “sal” should be salidroside? General procedures regarding the in vivo study must be included to clarify the protocol. Revise this section regarding typological errors; The keywords could be replaced by terms that were not used in the title and abstract.

Introduction – This section requires extensive revision. The problem is not clear, which hinders the comprehension of the study’s rationale. The authors should modify the order of the ideas, focusing on justifying the importance of conducting their work. I recognize that the hypothesis is worthy of investigation, however, the introduction is not clearly approaching the topics.

Please, provide additional data about salidroside, eg physicochemical properties;

What is NASH? (line 77)

The authors should remove the results summary provided at the end of the introduction section. Instead of mentioning the findings, you must emphasize the objectives. Please, revise it.

Material and methods – The preparation and characterization of QCOD@Sal are not clear. What was the total content of Sal in the final hydrogel? Did pH values determine?

The methodology of hemolysis was not presented. Please, revise it. Besides, concentration curves in both in vitro tests must be performed.

The authors must include more details regarding the ethical approval for animal experiments.

What is the difference between DNCB and DNFB? Figure 1a requires a correction; “DNFB infection” should be “DNFB exposure”. What is the green circle? It was not included in the caption.

What is the concentration of salidroside applied in the mice’s skin? The authors must cite proper references to justify the general procedures applied for in vivo experiments;

How many sets of experiments were performed? It is not clear to me if the same set of animals was used for ex vivo studies and NIR-II imaging. Please, clarify these issues and include the method used for euthanizing the animals.

The DPPH method should be mentioned in a separate section instead of mixed in the in vivo data.

The parameters evaluated in “Blood routine examination” should be listed in the methods section; The authors should mention how the data obtained were expressed;

The description of statistical evaluation is not adequate; The authors must cite the data normality assessment and the statistical test and post-test applied.

Results and discussion – The hydrogel seem to present non-Newtonian flow and Pseudoplastic behavior; The authors should better discuss such evidence.

The discussion regarding biocompatibility is shallow and must be complemented.

A proper statistical evaluation must be performed in the figure 4b;

The histological images are not optimal; The figures are too small and difficult to follow; The authors must improve their resolution and include proper identification of the inflammatory phenomena;

The conclusion could be shortened.

Author Response

Thank you for your suggestion, I have made the revision, please see the attachment.

Round 2

Reviewer 1 Report

The manuscript is acceptable for publication.

Author Response

Thank you very much for your comments.

Reviewer 2 Report

General comment:

Manuscript looks good, it is nicely improved, but it lacks few more informations.

According to the previous comments:

9)Please provide more details about staining

14)Question was how hydrogels improve skin hydration not what improve. Please describe it.

18) How the cell viability was calculated?

Additional comment:

Please provide the detail about hemolysis assay.

Author Response

Thank you for your insightful comments,Please see the attachment.

Reviewer 3 Report

I congratulate the authors on the rebuttal. The document was considerably improved. My previous concerns were almost totally solved, excepting the following:

a) The statistical analysis requires additional information; How were the data evaluated concerning statistical analysis (ANOVA or other) and post-test? Please, include such peace of detail in the revised manuscript.

b) The authors should remove the results summary from the introduction. I believe the focus must be on the objectives instead of the findings.

c) The discussion should be enhanced regarding the biocompatibility of the obtained formulation. The authors could link their findings with previous studies.

Author Response

(The authors gave the same response as above.)
